# Electrochemical Immunosensors for Quantification of Procalcitonin: Progress and Prospects

**Subramanian Nellaiappan** [1,2], **Pavan Kumar Mandali** [1,2], **Amrish Prabakaran** [1,2] and **Uma Maheswari Krishnan** [1,2,3,*]

1  Centre for Nanotechnology & Advanced Biomaterials (CeNTAB), SASTRA Deemed University, Thanjavur 613 401, India; nellaiappan.s@scbt.sastra.edu (S.N.); pavansmandali@gmail.com (P.K.M.); amrish1998@gmail.com (A.P.)
2  School of Chemical and Biotechnology, SASTRA Deemed University, Thanjavur 613 401, India
3  School of Arts, Science & Humanities, SASTRA Deemed University, Thanjavur 613 401, India
*  Correspondence: umakrishnan@sastra.edu; Tel.: +91-436-226-4101 (ext. 2677); Fax: +91-436-226-4120

**Abstract:** Human procalcitonin (PCT) is a peptide precursor of the calcium-regulating hormone calcitonin. Traditionally, PCT has been used as a biomarker for severe bacterial infections and sepsis. It has also been recently identified as a potential marker for COVID-19. Normally, serum PCT is intracellularly cleaved to calcitonin, which lowers the levels of PCT (<0.01 ng/mL). In severe infectious diseases and sepsis, serum PCT levels increase above 100 ng/mL in response to pro-inflammatory stimulation. Development of sensors for specific quantification of PCT has resulted in considerable improvement in the sensitivity, linear range and rapid response. Among the various sensing strategies, electrochemical platforms have been extensively investigated owing to their cost-effectiveness, ease of fabrication and portability. Sandwich-type electrochemical immunoassays based on the specific antigen–antibody interactions with an electrochemical transducer and use of nanointerfaces has augmented the electrochemical response of the sensors towards PCT. Identification of a superior combination of electrode material and nanointerface, and translation of the sensing platform into flexible and disposable substrates are under active investigation towards development of a point-of-care device for PCT detection. This review provides an overview of the existing detection strategies and limitations of PCT electrochemical immunosensors, and the emerging directions to address these lacunae.

**Keywords:** procalcitonin; electrochemical sensor; immunoassay; nanomaterials

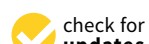



## 1. Introduction

Procalcitonin (PCT) is a 116-mer hormokine peptide formed by the neuroendocrine cells and the thyroid. It is cleaved to form the calcium-regulating hormone calcitonin. Its normal levels in healthy individuals are well below 0.01 µg/L. However, during infections, it is elevated and hence has emerged as a promising acute phase biomarker for the diagnosis of bacterial infections [1–4]. In addition, it has also been reported to serve as a marker for chronic obstructive pulmonary disorders, pneumonia, bronchitis, septic arthritis and medullary thyroid carcinoma [5]. Individuals affected with solid tumors who are more susceptible to infections could also be identified through monitoring their serum PCT levels [6] Recently, PCT has been found to be a marker for individuals infected with SARS-nCoV2 leading to COVID-19 [7–9]. Clinical studies have revealed that PCT levels correlate well with the severity of the disease [9]. An independent report has also suggested that elevated PCT levels in COVID-19 infected individuals are indicative of secondary bacterial infections [10]. In this context, the early and sensitive detection of PCT, an acute phase marker, may improve the prognosis for individuals affected by high-risk septicemia by aiding the design of appropriate therapeutic intervention. As most infections and inflammatory conditions involve high concentrations of several cytokines, highly specific

detection of PCT becomes important. Antibody-based sensing of PCT has been exclusively used for selective determination of PCT from samples. Such types of sensors are known as immunosensors. The most extensively investigated transduction mechanisms for quantification of PCT has been optical and electrochemical methods. The commonly employed strategy for PCT detection appears to be the usage of gold-labelled immune complexes for generating the electrochemical or optical response [4,5,11]. However, this method has several shortcomings, such as being semi-quantitative and expensive, shortage of gold labels, and poor sensitivity. Conventional immunoassays based on fluorescence (IFA), chemiluminescence (CLIA), and enzyme-linked immunosorbent assay (ELISA) have been employed for sensitive PCT detection but they are time-consuming, require larger sample volumes and are subject to interferences [5,12]. Moreover, they cannot be employed for real-time monitoring of individuals. Therefore, it is essential to develop a simple, rapid and sensitive detection strategy that offers a wide quantification range for PCT. Electrochemical immunosensors provide a versatile platform that can be tailored for desirable sensing range and sensitivity [13,14]. Further, the electrochemical sensing element can be translated in to disposable and point-of-care devices that enhance their utility for 'anywhere-anytime-anyone' use. The electrochemical immunosensor works on the principle in which the specific interaction between antibody and antigen is sensed by using a transducer and an electrical signal is measured at the modified electrode. The increase in thickness of the organic layer owing to the formation of immune complex results in an increase in the impedance or reduction in the current flowing through the system that is used as a measure of the amount of PCT in the sample. In particular, these specific interactions are used to detect immunochemical reactions either by direct or indirect methodologies [13,14]. In the indirect approach, sensing of the immune complex is achieved through labeling of either the antibody or the antigen with signaling molecules. Several immunoassay formats, such as sandwich type, competition, and capture, have been used (Figure 1).

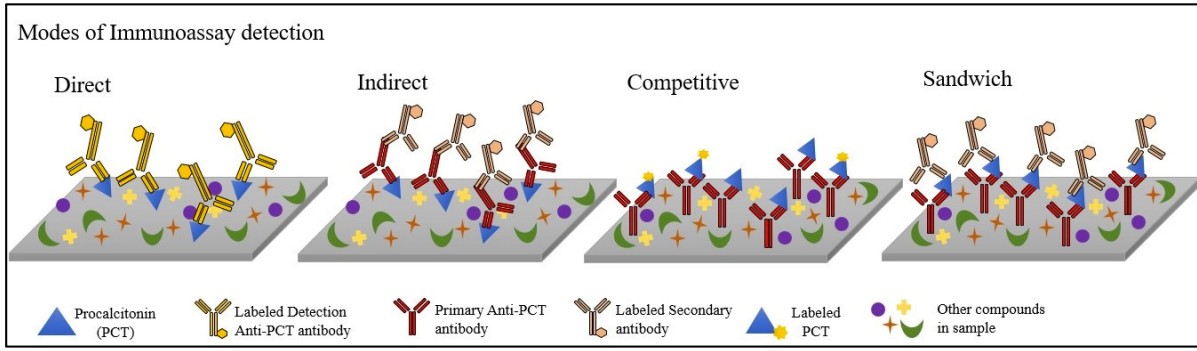

**Figure 1.** Schematic representation of various immunoassay formats.

In the sandwich or non-competitive assay, the antibody is immobilized on the sensor surface that serves to capture the antigen. Then a labelled secondary antibody which has affinity to another location of the antigen is added. The label is a fluorophore or dye in the case of optical sensors and generally, an enzyme in the case of electrochemical sensors. The intensity of the signal generated is correlated with the analyte in the sample. In the competitive assay, the analyte molecules compete with the labelled molecules for binding with the antibody. The intensity of the signal generated is indirectly proportional to the amount of analyte present in the sample. The capture assay is a direct immunoassay where the formation of the antibody–antigen immune complex is quantified through measuring the change produced in the property of the sensing element. Nanomaterials have elicited significant interest due to the considerable improvements in the analytical performance of electrochemical immunosensors they have brought about because of their unique physical and chemical properties, excellent conductivity and electrocatalytical activity [15,16]. The integration of nanostructures in the electrochemical sensing element have been found to

positively contribute to improved electron transfer and rapid response, as well as higher sensitivity. A wide range of nanomaterials of different shapes, sizes and combinations have been explored for electrochemical sensing [17].

Generally, the effective quantification of an analyte via electrochemical immunosensors involves two approaches, direct or indirect [14,18]. Although both methods have been explored for sensing PCT, the sandwich immunosensors have dominated the literature [19,20]. This is because the sandwich approach provides signal amplification and more specificity by the modification of selective and competitive counterpart systems. The use of an enzyme label like horseradish peroxidase on the secondary antibody results in the generation of additional electrons due to the redox reaction catalyzed by the enzyme label. In addition, electrochemiluminescence and photoelectroluminescence techniques have also been employed for quantification of PCT but the key to the success of these techniques lie on the development of a suitable luminophore and to overcome the complex steps involved in the immobilization of the antibody on the photoelectrode [19].

Various immunosensor platforms have been utilized, such as nanometals (gold, platinum, zinc) [21–33], metal oxides (cerium oxide, molybdenum oxide) [28–30], inorganic complexes (cobalt phthalocyanine) [33], carbon nanomaterials (graphene, reduced graphene oxide, multi-walled/single-walled carbon nanotubes, ordered mesoporous carbon, fullerene $C_{60}$) [21–26,33], highly branched polymers (poly(amidoamine), PAMAM) [34] with traditional redox partners (ferrocene, thionine and toluidine blue) [21–24,27,30,31] and quantum dots (zinc-sulfide-capped cadmium selenide) [35]. Both layer-by-layer modification or sandwich-type arrangement coupled with either labelled or label/enzyme-free immunoassays have been explored for detection of PCT. The present review presents an overview of the various electrochemical PCT immunosensors and nanointerfaces reported in available literature from past decade, i.e., 2012 to 2021. Emerging trends in detection methods have also been highlighted. The electrode modifications and performance of the existing electrochemical immunosensors are summarized and tabulated in Table 1.

**Table 1.** Comparison of the electroanalytical performances of various PCT electrochemical immunosensors available in the literature.

| Method | PCT Immunoelectrode (Signal Amplifier/Signal Tag) | Technique | Tag-Analytes | Linear Range | Detection Limit | Ref. |
|---|---|---|---|---|---|---|
| Sandwich-type | GCE/Graphene sheets/MWCNT/Chitosan/Glutaraldehyde/Ab$_1$/Bovine serum albumin/PCT/MCM/Thionine/AuNPs/HRP-Ab$_2$ | DPV | $H_2O_2$ | 0.01 to 350 ng/mL | 0.5 pg/mL | [21] |
| Sandwich-type | GCE/MWCNT/AuNPs/Ab$_1$/PCT/Glucose Oxidase@anti-PCT Ab$_2$-PtNPs-Fc-C$_{60}$ | DPV | $H_2O_2$ | 0.01 to 10 ng/mL | 6 pg/mL | [22] |
| Sandwich-type | GCE/rGO–Au/Ab$_1$/PTC/SWCNHs/HPtCs/HRP/Thionine–Ab$_2$ | DPV | $H_2O_2$ | 1 pg/mL to 20 ng/mL | 0.43 pg/mL | [23] |
| Label-free | Au/SWCNHs–PtNPs/PAMAM/Thionine–Ab$_1$/BSA/PCT | DPV | $H_2O_2$ | 10 pg/mL to 20 ng/mL | 1.74 pg/mL | [24] |
| Sandwich-type | GCE/Graphene oxide/Chitosan-Ab$_1$/PCT/Zn-OMCSi-Ab$_2$ | DPV | Zinc | 0.05 pg/mL to 80 ng/mL | 0.013 pg/mL | [25] |
| Sandwich-type | GCE/rGO-AuNPs/T-HRP/Ab$_1$/PCT/SA-HRP/Ab$_2$ | Amp $i$-$t$ | $H_2O_2$ | 0.05 to 100 ng/mL | 0.1 pg/mL | [26] |
| Sandwich-type | Au/PCT-Ab$_1$/PCT/Fc-AuNPs/PCT-Ab$_2$ | DPV | – | 1.5 pg/mL to 50 ng/mL | 0.8 pg/mL | [27] |
| Sandwich-type | GCE/AuNP/Ab$_1$/PCT/CuMn-CeO$_2$/Ab$_2$ | DPV | $H_2O_2$ | 0.1 pg/mL to 36.0 ng/mL | 0.03 pg/mL | [28] |
| Sandwich-type | GCE/AuNP/Ab$_1$/PCT/MoO$_3$/Au@rGO-Ab$_2$ | Amp $i$-$t$ | $H_2O_2$ | 0.01 pg/mL to 10 ng/mL | 0.002 pg/mL | [29] |
| Sandwich-type | GCE/CeO$_2$-CuO-Au/Ab/PCT/Au@Ag-Thionine-Ab$_2$ | SWV | – | 0.5 pg/mL to 50 ng/mL | 0.17 pg/mL | [30] |
| Enzyme-free | GCE/NiFe PBA nanocubes@Toluidene Blue/GA/PCT Ab/BSA/PCT | DPV | – | 0.001 to 25 ng/mL | $3 \times 10^{-4}$ ng/mL | [31] |
| Label-free | NiCo-MOF/MoS$_2$@PdNPs/CS/PCT-Ab | Amp $i$-$t$ | $H_2O_2$ | 0.001 to 50 ng/mL | 0.36 pg/mL | [32] |
| Sandwich-type | GCE/AuNP/Ab$_1$/BSA/Ag/nanoCoPC-MWCNTs/ChOx/Ab$_2$ | DPV | $H_2O_2$ | 0.01 to 100 ng/mL | 1.23 pg/mL | [33] |
| Enzyme-free | GCE/Fc-Fc/β-CD/PAMAM−AuNP/Ab$_2$ | DPV | Ascorbic acid | 1.80 pg/mL to 500 ng/mL | 0.36 pg/mL | [34] |
| Sandwich-type | AS-ITO/CdSeZnS-QD/PCT-Ab | CV (in [Fe(CN)$_6$]$^{3-}$) | – | 1 ng/mL to 10 µg/mL | 0.21 ng/mL | [35] |

## 2. Sandwich-Type Electrochemical Immunoassay

Most of the sensing strategies for PCT involve use of antibodies as the capture agent. This confers specificity to the analysis. However, immobilizing the antibody on the sensing element without compromising on its structural and functional stability remains a chal-

lenge. A diverse range of nanostructures have been explored as interface materials for the electrochemical detection of PCT. These are discussed in the following sections:

## 2.1. Carbon-Based Nanointerfaces

Carbon nanostructures have superior electron transport properties that aid electrochemical sensing. Different carbon nanostructures like multi-walled carbon nanotubes (MWCNTs), graphene, reduced graphene oxide (rGO), fullerenes have been used as interface materials. Fang et al. [21] developed a sandwich-type electrochemical immunosensor for PCT detection through layer-by-layer modification of the working electrode using a composite of graphene, carbon nanotubes, chitosan (GS/MWCNTs/CS) that also served as an immobilization surface for primary antibodies against PCT ($Ab_1$) through glutaraldehyde (GA) cross linker. The secondary PCT antibody ($Ab_2$), with horseradish peroxidase (HRP) label and gold nanoparticles coated with mesoporous silica conjugated through thionine linking (HRP-$Ab_2$/AuNPs/Thio/MCM41), served as an efficient platform for PCT electrochemical sensing in real human serum samples. Additionally, the HRP enzyme also catalyzed the electro-oxidation of thionine by $H_2O_2$ which resulted in an intense reduction peak in the presence of $H_2O_2$. The proposed GS/MWCNT/CS/GA/$Ab_1$/BSA/PCT/MCM/ Thio/AuNPs/HRP-$Ab_2$ immunosensor showed an impressive linear response from 0.01 to 350 ng/mL with a limit of detection 0.5 pg/mL (Figure 2). The sensor exhibited high specificity towards PCT and was retained about 88% of its current response even after 30 days of storage at 4 °C. The sensor displayed good precision and its values were in good agreement with those obtained using conventional ELISA indicating its potential to replace conventional assays for clinical diagnosis.

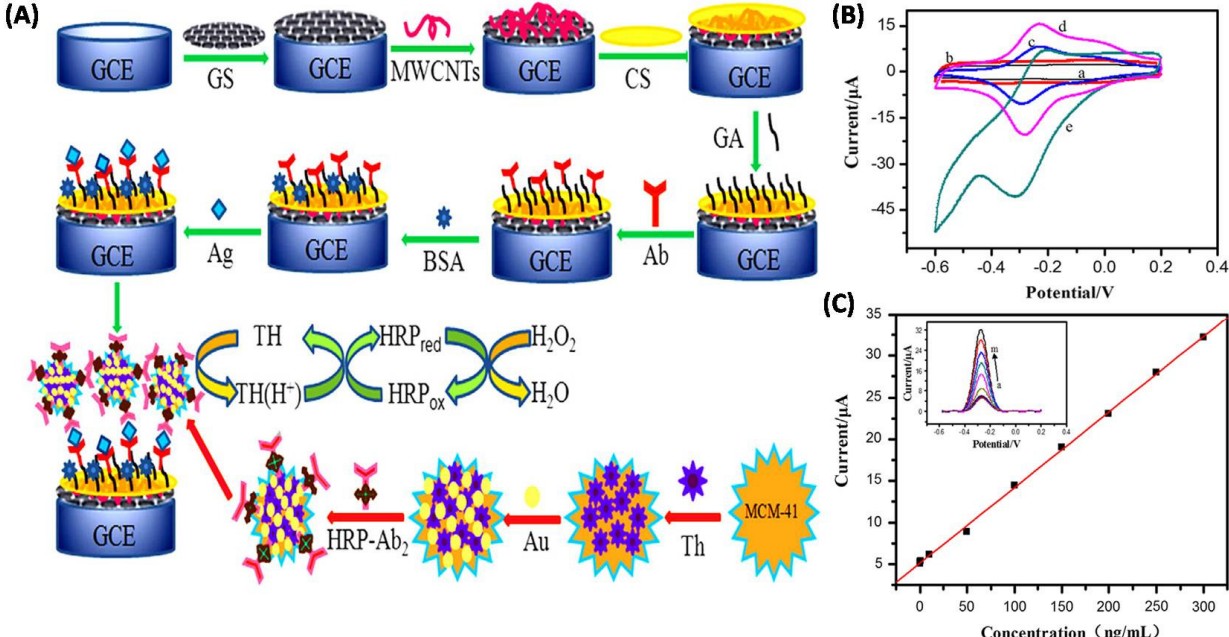

**Figure 2.** (**A**) Schematic depiction of the steps involved in the preparation of the immunosensor; (**B**) Cyclic voltammograms recorded at 50 mV/s for different electrode modifications: (a) $Ab_1$/GA/CS/MWCNTs/GS/GCE, (b) PCT/$Ab_1$/GA/CS/MWCNTs/GS/GCE, (c) HRP-$Ab_2$/Au/TH/MCM-41/PCT/$Ab_1$/GA/CS/GCE and (d) HRP-$Ab_2$/Au/TH/MCM-41/PCT/$Ab_1$/GA/CS/MWCNTs/GS/GCE in phosphate buffered saline of pH 7.0, and (e) HRP-$Ab_2$/Au/TH/MCM-41/PCT/$Ab_1$/GA/CS/MWCNTs/GS/GCE in medium containing 2 mM $H_2O_2$; (**C**) Calibration plot obtained for the electrochemical immunosensor using known concentrations of PCT in PBS of pH 7.0 containing 2 mM $H_2O_2$, *n* = 5 for every concentration level in the same measurement run [21].

In another sandwich-type immunosensor [22], MWCNT functionalized with gold nanoparticles (AuNPs) served as an active immobilization surface for the primary anti-

PCT antibody (MWCNT/AuNPs/PCT-Ab$_1$) through spontaneous chemical interactions between AuNPs and thiol groups present in the antibodies. Post capture of the PCT antigen by the primary antibody, the secondary PCT antibody (Ab$_2$) labeled with glucose oxidase (GO$x$-labelled-PCT Ab$_2$) was linked to ferrocene (Fc) carboxylic acid redox probe conjugated to amino-functionalized fullerene C$_{60}$ on platinum nanoparticles (PtNPs-Fc-C$_{60}$ nanocomposite) to form an immune-analyte complex. The efficiency of the MWCNT/AuNPs/Ab$_1$/PCT/GO$x$@anti-Ab$_2$/PtNPs-Fc-C$_{60}$ immunosensor arises due to amplification arising out of the synergistic redox capabilities of Fc molecule, oxidation of glucose to H$_2$O$_2$ by GO$x$ enzyme and the subsequent reduction of H$_2$O$_2$ by Pt-containing nanocomposite.

Recently, an ultrasensitive sandwich electrochemical immunosensor was developed using single-walled carbon nanohorns (SWCNHs) in combination with hollow platinum chain complex (SWCNHs/HPtCs) for PCT detection in real clinical serum samples [23,24]. The sensor device also employed a composite interface film of reduced graphene oxide (rGO) and gold nanoparticles for immobilization of the primary antibody against PCT (rGO-Au/PCT-Ab$_1$). The SWCNHs/HPtCs was conjugated to the secondary antibody labeled with HRP enzyme and the redox mediator thionine (SWCNHs/HPtC/HRP/Thio/PCT-Ab$_2$) to obtain an amplified current response [23]. The proposed immunosensor SWCNH/HPtC/HRP/thi–Ab$_2$/PTC/Ab$_1$/rGO–Au showed good responses with linearity from 1 pg/mL to 20 ng/mL of PCT with a detection limit of 0.43 pg/mL. The HPtCs served as a biocompatible matrix for immobilization of HRP and antibodies as well as catalyzed the reduction of H$_2$O$_2$ in a display of a synergistic effect with the enzyme. In a related work, the SWCNHs/HPtCs complex was chemically conjugated with the hyperbranched polymer PAMAM (G4.0) and the primary antibody (PCT-Ab$_1$). The electrochemical response was obtained after addition of the redox probe thionine and HRP in the presence of H$_2$O$_2$ [24]. This study employed only the primary Ab$_1$ as the capture element for the detection of PCT without the secondary antibody (PCT-Ab$_2$) unlike earlier reports in the literature [23]. The linear response for this sensor was observed from 10 pg/mL to 20 ng/mL with a detection limit value of 1.74 pg/mL, which was however, four times lower than a similar interface reported earlier by the same group employing a secondary antibody.

Feng et al. [25] designed a system comprising zinc nanoparticles decorated ordered mesoporous carbon silica nanocomposites with detection antibodies (Zn-OMCSi-Ab$_2$) for ultrasensitive quantification of PCT in human serum samples. A combination of reduced graphene oxide, chitosan and glutaraldehyde was used to immobilize primary antibody through covalent linkage of the amino groups of chitosan (rGO/CS-Ab$_1$). The sensing performance of the immunoelectrode (GCE/rGO/CS-Ab$_1$/PCT/Zn-OMCSi-Ab$_2$) towards PCT was attributed to the direct oxidation of the entrapped zinc nanoparticles after formation of the sandwich-type immunoreactions on the electrode surface. The oxidation current exhibited good linear correlation for PCT concentrations from 0.05 pg/mL to 80 ng/mL, with a limit of detection 0.013 pg/mL (Figure 3).

Recently, a PCT sensor based on gold nanoparticles-tyramide-labeled biotinylated HRP (AuNPs/T-HRP) and a nanocomposite of reduced graphene oxide nanosheets-gold nanoparticles with primary antibodies (rGO-AuNPs/T-HRP/Ab$_1$) was fabricated [26]. Another nanocomposite of HRP-streptavidin conjugated with the secondary antibody (SA-HRP/Ab$_2$) generated the electrochemical response. The high affinity interactions between streptavidin and biotin resulted in amplified current signals. The use of gold nanoparticles enabled immobilization of greater number of HRP on its surface that contributed to the high sensitivity of the sensor. HRP in the presence of peroxide oxidized tryramide to quinone, which caused a further amplification of the signal. This GCE/rGO-AuNPs/HRP/Ab$_1$/PCT/SA-HRP/Ab$_2$ immunoelectrode detected PCT between 0.05 ng/mL and 100 ng/mL with an ultralow detection limit of 0.1 pg/mL (Figure 4).

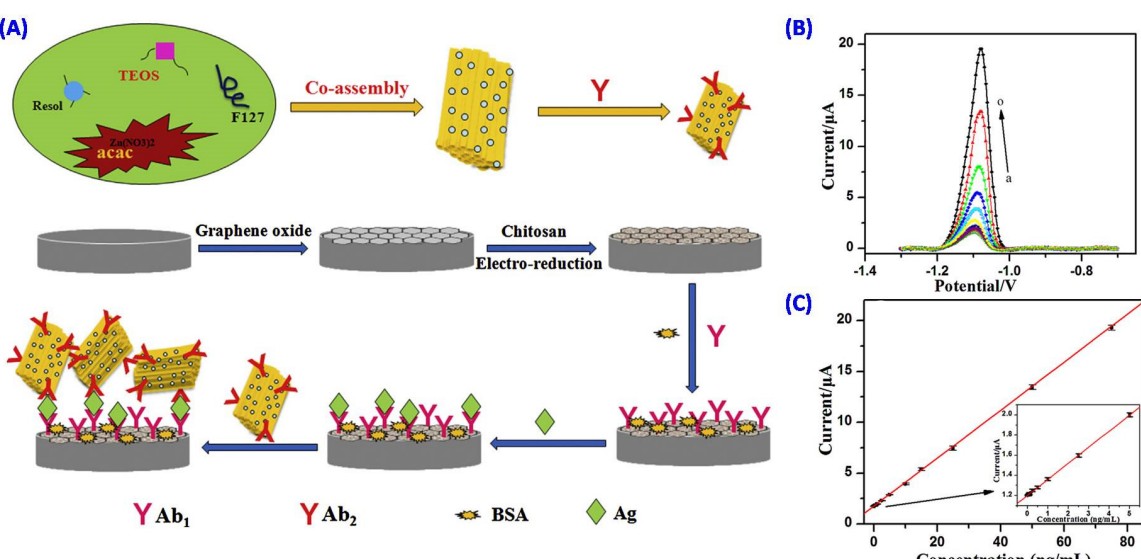

**Figure 3.** (**A**) Schematic representation of the stages involved in the preparation of Ab$_2$-OMCSi-Zn bioconjugate and the sandwich-type immunosensor; (**B**) DPV recorded for the immunosensor towards various PCT concentrations (0.005, 0.01, 0.05, 0.075, 0.1, 0.25, 0.5, 1, 2.5, 5, 10, 15, 25, 50, 75 ng/mL) in pH 6.0 acetate buffer. (**C**) Calibration plot obtained for PCT using the immunosensor [25].

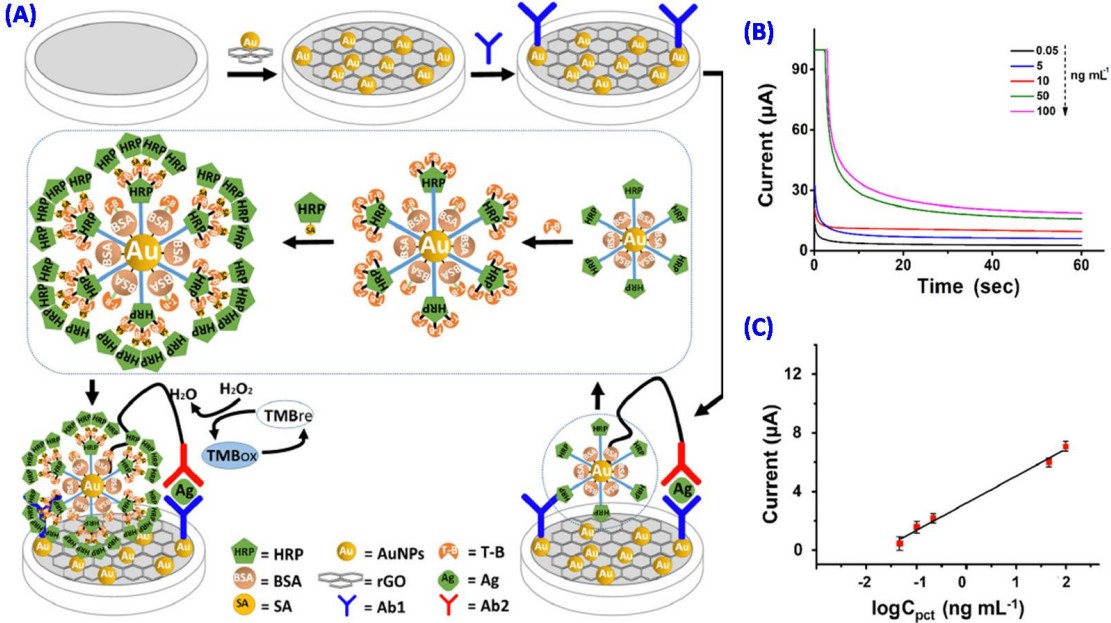

**Figure 4.** (**A**) Schematic depiction of the step-wise preparation of the PCT immunosensor; (**B**) Amperometric profiles of the immunosensor in different concentrations between 0.05 and 100 ng/mL of PCT recorded in 0.01 M PBS of pH 7.6 containing 0.5% bovine serum albumin. (**C**) The calibration plot obtained for the immunosensor [26].

Graphitic carbon nitride interface containing homogenously dispersed nickel cobalt-sulphide (NiCo$_2$S$_4$) and silver nanoparticle-entrapped MWCNTs served as a hybrid interface for immobilizing anti-PCT antibodies [36]. The silver nanoparticle containing MWCNTs facilitated electron transfer and augmented the signal from differential pulse voltammetry (DPV) while the NiCo$_2$S$_4$ served as a bimetallic indicator for chronoamperometery. The sensor detected PCT between 0.05 to 50 ng/mL with a detection limit of 16.7 pg/mL in the DPV mode and 1 pg/mL to 10 ng/mL with a limit of detection 0.33 pg/mL in the chronoamperometry mode. The high sensitivity and stability of this

sensor could be explored further in a clinical setup to understand its utility for routine clinical diagnosis. While graphitic carbon nitride served as a matrix in this strategy, it was employed as a catalyst in another sensing device designed for PCT quantification. Glassy carbon electrode was modified with the two-dimensional titanium carbide MXene layer [37]. The MXene was doped with sulfur to favor incorporation of gold nanoparticles and the anti-PCT antibody. The sensing strategy involved the addition of graphitic carbon nitride linked with the secondary antibody after incubation with the analyte. The graphitic carbon nitride served as a redox catalyst for generating the electrochemical response from $H_2O_2$. This sandwich immunosensor exhibited a linear range of 0.01–1 pg/mL with an ultralow detection limit of 2 fg/mL and response time of 12 s. However, the sensor will benefit from additional optimization for extending its linear range for better clinical translation.

One of the key challenges in immunosensors is maintenance of structural, functional stability and accessibility to the binding site of the antibody, which influences the sensitivity and sensing range. In a novel strategy to orient the antibody and improve the sensitivity of an electrochemiluminescent sensor designed for quantification of PCT, a heptameric oligopeptide with the sequence HWRGWVC was introduced over a nanocomposite layer comprising gold nanoparticle dispersed reduced graphene oxide functionalized with poly(aniline) nanorods. The peptide served to immobilize and orient the anti-PCT antibody [38]. The iron storage protein ferritin conjugated to the electrochemiluminescent probe N-(aminobutyl)-N-(ethylisoluminol) (ABEI-Ft) and secondary antibody served to generate the optical signal in response to the captured PCT. This approach resulted in an astonishing detection limit and linear range of 54 fg/mL and 100 fg/mL–50 ng/mL, respectively. This concept can be extended to a purely electrochemical platform to achieve highly sensitive detection of PCT.

### 2.2. Nanometallic and Metal Oxide-Based Interfaces

Metallic nanoparticles in combination with redox probes have been extensively explored as interface materials in sandwich-type electrochemical immunosensors for the detection of PCT. Ferrocene-modified gold nanoparticles labeled with secondary PCT antibody (Fc-AuNPs/PCT-$Ab_2$) have been used for electrochemical signal amplification when it binds to the immune complex formed between PCT and the primary antibody modified gold disk electrode (Au/PCT-$Ab_1$) to form a sandwich structure for the determination of PCT in clinical samples [27]. The sensor showed good linearity between 1.5 pg/mL and 50 ng/mL with a detection limit of 0.8 pg/mL. Apart from acting as redox-mediators, nanometals and metal oxides have also been used as redox probes and nanocatalysts. For instance, simultaneous doping of copper and manganese into ceria (CuMn-$CeO_2$) nanocomposite in combination with secondary $Ab_2$ has been reported for the sensitive immunoassay of PCT [28]. The double-doping introduced additional oxygen vacancies into the $CeO_2$ lattice thereby enhancing the redox and catalytic activities of the interface towards $H_2O_2$ for signal amplification. In addition, it increased the immobilization of $Ab_2$ through chemical interactions between carboxylic groups of $Ab_2$ and $CeO_2$ through formation of ester-like bridges. The constructed GCE/AuNP/$Ab_1$/PCT/CuMn-$CeO_2$/$Ab_2$ immunosensor exhibited a wide linear range of 0.1 pg/mL to 36.0 ng/mL for PCT with a low detection limit of 0.03 pg/mL.

Another study employed $CeO_2$-CuO-Au catalytic nanointerface (Figure 5) to immobilize the primary antibody against PCT while a heterojunction of gold and silver nanoparticles containing thionine redox mediator, and the secondary antibody served as the electrochemical probe [30]. The sensor detected PCT concentrations between 0.5 pg/mL and 50 ng/mL in simulated samples as well as in serum. A similar strategy was explored using $Fe_3S_4$/Pd nanocomposite as the catalytic interface over glassy carbon electrode [39]. The magnetic interface adhered well to the electrode and also served to immobilize the anti-PCT antibody. The secondary antibody was conjugated over mesoporous bioactive glass using glutaraldehyde cross-linking reaction between the amino groups in the meso-

porous matrix and antibody. The sensor detected PCT with high specificity in the linear range of 500 fg/mL to 50 ng/mL. The sensor also performed well when employed to detect PCT in spiked serum samples. The use of the insulating mesoporous matrix was attributed to the sharp decrease in the current resulting in high sensitivity. A similar concept of using high surface area mesoporous matrix was reported using gold decorated mesoporous silica matrix loaded with the redox mediator thionine that was coated over glassy carbon electrode [40]. This strategy avoids the use of a labeled secondary antibody and hence could provide a cost-effective option for quantification of PCT. The high surface area silica matrix also served to immobilize PCT antibody. The sensor displayed good linearity between 0.001 and 100 ng/mL of PCT when probed using differential pulse voltammetry. Mesoporous silica incorporated with silver nanoparticles and toluidine blue with covalently linked secondary antibody was also investigated as the signal enhancer in an electrochemical sensor where the primary antibody was immobilized on a glassy carbon electrode coated with a nanointerface comprising PtCoIr nanowires modified with polyethylenepolyamine-linked ferrocene [41]. A linear sensing range of 0.001–100 ng/mL of PCT was achieved and the sensor also performed well in serum samples. In another immunosensor, hybrid molybdenum oxide and gold nanoparticles decorated reduced graphene oxide combined with secondary antibody ($MoO_3$/Au@rGO-$Ab_2$) has developed as the signal amplification material for PCT detection [29]. The synergistic effect of this $MoO_3$/Au@rGO hybrid nanocomposite exhibited excellent electrocatalysis of $H_2O_2$ towards low concentration of PCT. The fabricated GCE/AuNP/$Ab_1$/PCT/$MoO_3$/Au@rGO-$Ab_2$ showed a wide working range from 0.01 pg/mL to 10 ng/mL with detection limit value of 0.002 pg/mL.

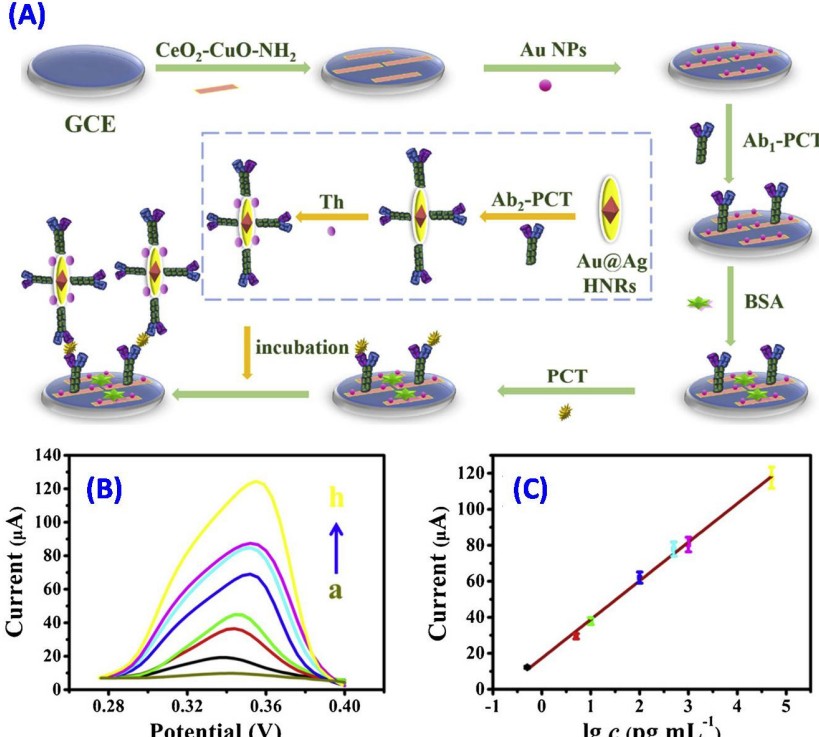

**Figure 5.** (**A**) Schematic representation of the sandwich-type electrochemical immunosensor with $CeO_2$-CuO-Au nanointerface. (**B**) Square wave voltammetry current response for different concentrations of PCT (From a–h: 0, 0.5, 5, 10, 100, 500 pg/mL and 1, 50 ng/mL), (**C**) Linear calibration plot obtained for the immunosensor towards quantification of different concentrations of PCT. Error bar represents standard deviation (*n* = 5) [30].

Another interesting combination of interface materials involved the use of photoactive $NiTiO_3$ nanorods decorated with $Zn_xBi_2S_{3+x}$ nanoparticles distributed homogenously over indium tin oxide (ITO substrate) [42]. The hybrid nanointerface served as an immobi-

lization layer for the anti-PCT antibody, and as a photoelectrocatalyst that generated the electrochemical signal through oxidation of glucose. This versatile system exhibited a wide linear range of 0.0001 to 50 ng/mL PCT with high reproducibility. It also performed well in spiked serum samples, indicating its promise for clinical applications. A similar sensing strategy was employed for quantification of PCT using CdS-Bi$_2$Sn$_2$O$_7$-anti-PCT complex as the photoelectrocatalyst and PCT capture matrix [43]. The electrochemical signal for this immunosensor was generated by the redox reaction involving thiocholine catalyzed by the acetylcholinesterase enzyme linked to silica nanospheres. This highly sensitive system quantified PCT effectively between 0.0005 to 100 ng/mL.

### 2.3. Inorganic Metallic Interfaces and Organic Framework-Based Immunosensors

When compared to metallic and metal oxide interfaces, inorganic and metal-organic frameworks have been less explored as interfaces in PCT sensors. However, growing evidence of the improved sensing characteristics conferred by these structures has triggered research towards harnessing the sensing characteristics of these three-dimensional framework-based composites as label-free or enzyme-free platform in electrochemical immunosensors [31,32]. An enzyme-free electrochemical immunosensor based on toluidine blue redox probe and self-templating NiFe-prussian blue nanocubes (NiFe-PBA) functionalized with anti-PCT antibody using glutaraldehyde cross-linker has been reported for detection of PCT [31,32]. The synergetic effect between the redox mediators toluidine blue and NiFe PBA nanocubes enabled excellent signal amplification without any additional biorecognition elements. The sensor displayed a wide detection range from 0.001 to 25 ng/mL for PCT with a low detection limit of $3 \times 10^{-4}$ ng/mL. In another approach, transition metal dichalcogenides in combination with cubic metal-organic framework (MoS$_2$/NiCo-MOF) heterostructures were fabricated for sensitive label-free determination of PCT [32]. These heterostructures were further functionalized with in situ synthesized palladium nanoparticles (Pd NPs) for additional catalytic activity towards the reduction of H$_2$O$_2$. The Pd NPs@MoS$_2$/NiCo-MOF matrix provided a large surface area for immobilization of the primary antibody upon incubation with chitosan. The sensor exhibited a linear range of 0.001–50 ng/mL with a detection limit of 0.36 pg/mL towards PCT quantification (Figure 6).

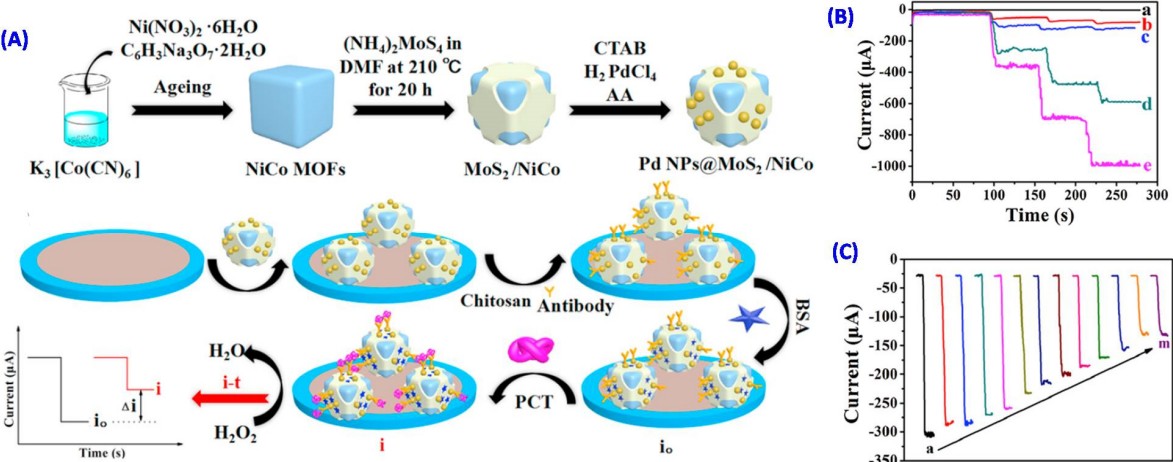

**Figure 6.** (**A**) Representation of the steps involved in the fabrication of Pd NPs@MoS$_2$/NiCo heterostructures and (**B**) Fabrication of the immunosensor, chronoamperometric detection of PCT. (**B**) Chronoamperometric curves obtained for different electrodes at −0.1 V in PBS of 0.07 mol/L concentration and pH 7.38, for three consecutive additions of 0.015 mol/L H$_2$O$_2$: (a) NiCo MOFs, (b) MoS$_2$/NiCo, (c) Pd NPs@NiCo, (d) Pd NPs@MoS$_2$ and (e) Pd NPs@MoS$_2$/NiCo. (**C**) Chronoamperometric responses recorded for the immunosensor upon addition of different concentrations of PCT (from a to m: 0, 0.0005, 0.001, 0.005, 0.01, 0.05, 0.1, 0.5, 1, 5, 10, 50, 100 ng/mL) in 10 mL PBS (0.07 mol/L, pH 7.38) containing 5.0 mmol/L H$_2$O$_2$ [32].

In an interesting strategy, glassy carbon electrode was modified with silica matrix doped with gold nanoparticles for improved conductivity [44]. The gold-doped silica matrix was used to link the electrochemical signal probe ferrocene carboxylic acid and the primary antibody against PCT. The metal organic framework (MOF) UiO-66 containing $Zr_6O_4(OH)_4$ linked using 1,4-benzenedicarboxylate was used to electrostatically retain the anionic redox mediator toluidine blue, as well as serving to covalently immobilize the secondary antibody. The ratio of the redox signals from the ferrocene carboxylic acid and toluidine blue during DPV was used to quantify PCT between the wide concentration range of 1 pg/mL and 100 ng/mL with a detection limit of 0.3 pg/mL. The sensor was validated using spiked serum samples and can be explored for clinical applications.

### 2.4. Metal Complex-Based Immunosensors

Recently, label-free sensors have gained attention for improvement in the analytical speed and reduction in complexity. Metal complexes have emerged as a promising candidate for label-free immunosensors. In a typical example, cobalt phthalocyanine (CoPC) nanoparticles decorated MWCNTs immobilized with primary $Ab_1$ against PCT on gold nanoparticle-layered glassy carbon electrode have been developed for sensitive detection of PCT in clinical samples [33]. The redox property of CoPC occurs via the Co(II)/Co(I) redox couple and could enhance the electrochemical signal without addition of other redox mediators. A cascade signal amplification system has been attempted for this system by chemical conjugation of choline oxidase (ChO*x*) and a secondary $Ab_2$ immobilized on CoPC/MWCNT to catalyze the conversion of choline substrate to $H_2O_2$ that produces the electrochemical response. This system exhibited linearity for PCT concentrations between 0.01 and 100 ng/mL with a low detection limit of 1.23 pg/mL towards PCT detection (Figure 7).

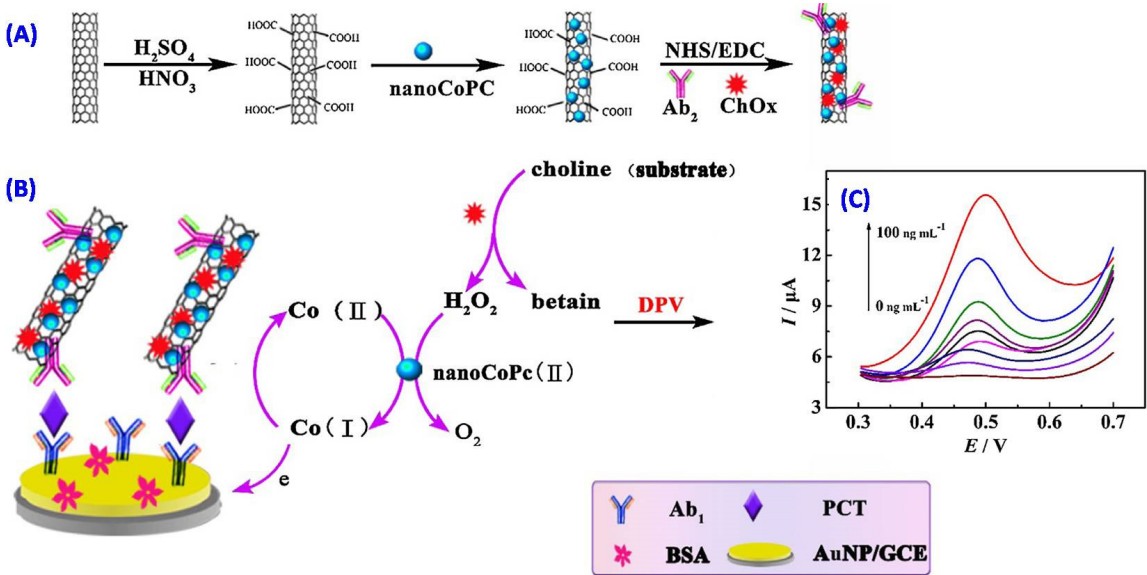

**Figure 7.** (**A**) Preparation steps involved in the fabrication of ChOx/$Ab_2$/nanoCoPc-MWCNTs bioconjugates. (**B**) Fabrication process of electrochemical immunosensor and the amplification mechanism of detection signal. (**C**) DPV responses recorded for the immunosensor after incubating with different concentrations of PCT from 0.01 to 100 ng/mL under optimized conditions [33].

### 2.5. Host–Guest Interfaces in Immunosensors

In a seminal work, a novel host-guest molecular recognition approach has been successfully employed for the development of an enzyme-free electrochemical immunosensor using N,N-bis(ferrocenoyl)-diaminoethane/β-cyclodextrin/poly(amidoamine) dendrimer–Au nanoparticle composite (Fc-Fc/β-CD/PAMAM-Au) immobilizing the secondary anti-

body (PCT-Ab$_2$) for PCT detection. The catalysis of ascorbic acid after the binding of the secondary antibody to PCT generated the electrochemical signal [34]. The Fc-Fc facilitated extensive linking with β-CD/PAMAM-Au through its electroactive groups to form a net-like nanostructure. The PAMAM-Au conjugate had dual roles of serving as a nanocarrier for increased immobilization of the Ab$_2$ and β-cyclodextrin, and acted as nanocatalyst for efficient electro-oxidation of ascorbic acid that served to amplify the electrochemical response. The immunosensor exhibited a low detection limit (0.36 pg/mL) and a broad linear range of 1.80 pg/mL to 500 ng/mL for PCT detection. However, further validation with clinical cohorts is necessary to realize its potential for disease diagnosis.

### 2.6. Quantum-Dots-Based Immunosensors

Quantum dots are semi-conductor nanoparticles with a tunable band gap that has been explored for sensing applications. Recently, water-soluble core/shell quantum dots have been employed as a functional interface material in electrochemical sensors, biosensors and immunosensors due to its active participation in the electron, as well as charge transfer reactions. For instance, a self-assembled system comprising of amine-functionalized and aqueous soluble zinc-sulphide-capped cadmium selenide core-shell quantum dots (CdSeZnS-QD) was conjugated with anti-PCT antibody on aminosilane modified indium-tin oxide (ITO)-coated glass substrate using glutaraldehyde (GA) cross-linking. This electrochemical immunoelectrode (AS-ITO/CdSeZnS-QD/PCT-Ab) was successfully employed for the quantification of PCT for diagnosis of urinary tract infection (UTI) [35]. The quantum dots served as charge transfer agents that increased the electroactive surface area and improved the electron transfer kinetics of the immunoelectrode. This sensor detected PCT in the concentration range from 1 ng/mL to 10 mg/mL with a detection limit value of 0.21 ng/mL. The removal of the quantum dots from the electrode surface resulted in a steep decrease in the sensitivity of the sensor which now had a linear range between 0.1 and 10 mg/mL. This study highlights the importance of the nano-interface towards sensitive detection of PCT in the range reported for the specified infection.

### 2.6.1. Other Electrochemical Detection Strategies for PCT

Most of the electrochemical sensing platforms for PCT have employed anti-PCT antibodies for specific detection. However, antibodies are high-molecular-weight proteins that are susceptible to loss of functionality due to denaturation, and are expensive. Attempts to replace the antibodies by low-molecular-weight agents without compromising on the substrate specificity have resulted in the development of peptides identified using phage display. A dodecapeptide sequence MSCAGHMCTRFV derived from phage display was found to exhibit high binding affinity to PCT [45]. In a related work, four peptide sequences derived from phage display were immobilized over gold working electrodes through thiol linkers and their electrochemical response to PCT was monitored using electrochemical impedance spectroscopy [46]. Electrodes modified with the peptide showing highest binding affinity to PCT displayed a linear range of 0.0125–0.25 μg/mL. This strategy opens up newer options for development of the sensing element towards PCT. However, it requires additional fine-tuning to improve the sensing range, especially in the lower concentrations.

Conventional sandwich assays employ an enzyme label to generate the electrochemical signal in response to the analyte. In an interesting variant, the use of the enzyme label was substituted by glucose-encapsulated liposomes that were conjugated with the detection antibody [47]. After the PCT in the sample bound to the immobilized primary antibody on the electrode surface, the secondary antibody was added. This was followed by addition of the detection antibody-tagged glucose encapsulated liposome. The electrochemical signal was generated by oxidation of glucose released upon lysis of the liposome by the surfactant Triton-X 100, using a conventional glucometer. This sensor reported a detection limit of 0.15 nM for PCT, which corresponds to 0.52 ng/mL and a detection range between 0.153–15.38 nM. Further experiments to validate the performance of the sensor using clinical samples are required for bench-to-bed translation.

### 2.6.2. Multiplexed Sensors

The accuracy of clinical diagnosis of a condition can be improved through quantification of multiple markers. A host of inflammatory markers are deregulated during sepsis and infections. Hence, attempts to develop a device containing an array of sensing elements that can simultaneously detect multiple markers are actively being pursued. Several recent reports on multiplexed sensors with PCT as one of the analytes are available in literature [48]. A lab-on-a-chip immunosensor array was reported for simultaneous detection of the pro-inflammatory markers PCT, interleukin-6 (IL-6) and C-reactive protein (CRP) using a cyclooctene (TCO)-tetrazine (Tz)-horse radish peroxidase (HRP) assembly [49]. The ratio of TCO-HRP and Tz-HRP was optimized to achieve adequate chemiluminescent response for the different markers that are present in varying levels in blood. The linear ranges obtained for CRP, PCT and IL-6 were 0.2–100 µg/mL, 0.14–700 ng/mL and 0.025–500 ng/mL, respectively. The device was validated using spiked samples and could be used as a diagnostic device for inflammatory conditions. In a seminal work, three key biomarkers of sepsis, namely, PCT, CRP and immobilized Fc-mannose binding lectins (Fc-MBL) displaying pathogen associated molecular patterns (PAMP) were simultaneously detected from whole blood employing a gold electrode coated with a nanocomposite interface comprising reduced graphene oxide, glutaraldehyde cross-linked bovine serum albumin and tetramethyl benzidene [50]. The multiplexed sensor exhibited a linear range of 1.06–48.8 ng/mL for PCT, 0.63–3.76 µg/mL for CRP and a detection limit of 6 ng/mL for the Fc-MBL PAMPs. No cross-reactivity between the analytes was reported. A proof-of-concept work involving a flexible polyimide substrate with interdigitated gold electrodes coated with 100 nm thick zinc oxide film was employed for simultaneous detection of PCT and CRP. The respective antibodies were immobilized on the electrodes through dithiobis (succinimidyl propionate) linker. The point-of-care device employed electrochemical impedance spectroscopy (EIS) to quantify PCT between 0.01–10 ng/mL and CRP between 0.01–20 µg/mL with excellent specificity using 10 µL of sample. This device could serve as a point-of-care device for diagnosis of sepsis. In another effort at detection of sepsis, a panel of three biomarkers, namely, PCT, lipopolysaccharide (LPS) and lipoteichoic acid (LTA), were quantified simultaneously using a fabricated microfluidic device with nanochannels [51]. Gold microelectrodes printed on a substrate and PDMS layer with nanochannels was positioned through pressure adhesion over the sensing area. A nylon membrane interlayer modified with the antibodies against the respective analytes was placed over the gold electrodes. The device exhibited a linear response of 0.1 ng/mL–10 µg/mL for PCT and 1–1000 µg/mL for LPS and LTA. The total response time for the analysis was 15 min. Further studies in a disease cohort will serve to realize the clinical potential of this device. In a recent study, a disposable screen-printed electrode platform was designed for simultaneous detection of PCT and CRP from neonatal blood samples [52]. The sensing strategy was based on sandwich immunoassay. The novelty was the use of magnetic beads functionalized with the capture antibody that retained the PCT on the electrode surface in the presence of an external magnetic field. The addition of the secondary antibody labeled with horseradish peroxidase resulted in the generation of the electrochemical signal that was recorded using amperometry. The sensor detected PCT between 0.25 and 100 ng/mL, and CRP between 0.01–5 µg/mL. The sensor was used to detect PCT levels from serum samples of healthy and infected infants with good correlation and accuracy when compared with conventional method. Apart from sepsis, PCT has also been introduced in a biomarker panel for diagnosis of cardiovascular disorders. A paper-based point-of-care device was fabricated for detection of cardiac disease markers PCT, cardiac troponin I (cTnI) and CRP. Antibodies were used as the capture agent for the individual analyte and were immobilized over the stencil-printed carbon electrodes coated with graphene oxide [53]. Square wave voltammetry measurements revealed a linear response for CRP between 0.001 and 100 µg/mL, 0.5 pg/mL–250 ng/mL and 0.001 to 250 ng/mL for PCT and cTnI, respectively. Apart from specificity, the device exhibited stability for about a month.

This device could have immense utility as an early diagnostic system for cardiovascular diseases.

### 2.6.3. Emerging Directions

Microfluidic chips and low-cost disposable electrochemical platforms with low sample volume are emerging as the front-runners in clinical diagnosis. A disposable screen-printed platform and a microfluidic chip were both explored for PCT detection using antibody labeled magnetic beads for capture and horseradish peroxidase labeled secondary antibody for generating the electrochemical signal. Both platforms resulted in a rapid response within 20 min over a wide range of PCT concentrations ranging from 0.5–1000 ng/mL using 25 μL of the sample. Along similar lines, a recent study demonstrated the use of a polydimethyl siloxane (PDMS) microfluidic chip containing gold electrodes that could simultaneously detect PCT and IL-6 for early diagnosis of sepsis from ultra-low sample volume of 10 μL [54]. The sensing was accomplished using electrical counting at the entrance and the exit of the flow channel. The method employed different sized microbeads linked with the capture antibodies for PCT and IL-6. The size differences in the microbeads for each marker resulted in different pulse frequencies and amplitudes that were used to discriminate the counts for PCT and IL-6. The flow channels contained secondary antibodies for the respective markers conjugated using avidin–biotin linkage. The beads were again counted at the exit and the difference was used to quantify the amount of PCT and IL-6 in the samples. This device exhibited a detection limit of 130 and 150 pg/mL for PCT and IL-6, respectively. The sensor also was validated using spiked plasma samples. This platform could be further expanded for additional markers using different sized microbeads for more precise diagnosis. An electrochemical magneto-immunoassay platform was successfully fabricated using gold electrode in a microfluidic chamber that employed an ultra-low volume of 25 μL. Streptavidin-conjugated magnetic beads were used to retain biotinylated anti-PCT antibody on their surface. A neodymium magnet placed beneath the chamber kept the magnetic beads in position. Horseradish peroxidase conjugated secondary antibody was used to generate the electrochemical signal. The microfluidic sensor had a linear range between 0.5 and 100 ng/mL of PCT with a detection limit of 0.02 ng/mL. The detection of PCT from pre-term neonatal serum samples was achieved in 15 min, making this a promising tool for rapid clinical diagnosis of sepsis [20]. Integration of machine learning and deep learning algorithms may also be explored in tandem with the device to improve the precision and accuracy of discrimination and quantification of the biomarker panels. Another emerging vista in the design of sensing platforms for clinical biomarkers is the fabrication of portable devices using 3D printing. A recent approach had employed 3D printing to design the mechanical components of a portable sensor platform that used optical cavity absorption for sensitive quantification of PCT, CRP and IL-6 [55]. This design strategy could well be explored for fabricating electrochemical sensing chips in the near future.

Aptamers represent an interesting class of short oligonucleotide sequences that display specific binding to a specific target similar to antibodies. The use of aptamer sequences overcomes problems such as denaturation, variations during affinity maturation and cost that are encountered with use of antibodies. Aptamers for a specific target are identified through high-throughput screening. Aptasensors have been explored for several clinical markers like eosinophil cationic protein [56] but have been less explored for procalcitonin. An attempt to fabricate an aptasensor for PCT resulted in poor sensitivity and linear range [57]. Nevertheless, the key lies in the design of the aptamer sequence.

## 3. Concluding Remarks and Future Perspectives

Sensors have made a mark in all domains of life and continuous efforts to improve the sensing performance have spurred the search for efficient materials for sensing or transduction. The growing importance of PCT as a disease biomarker independently, as well as in combination with other disease markers has now opened up new avenues for

the development of effective electrochemical sensor platforms for affordable commercial devices for clinical diagnosis. In this review, we have summarized the available electrochemical immunosensors for sensitive and selective recognition of PCT in clinical serum samples. The nanointerfaced immunosensors exhibit quick responses, excellent stability, high sensitivity and selectivity, and in most cases have been found to be comparable to the conventional ELISA method. However, use of antibodies as the capture element makes the technology expensive. Moreover, variations in the binding affinity of the commercially available antibodies towards the antigen during affinity maturation reduces reproducibility. Aptamers that exhibit specific binding to the targets may be an effective alternative for capturing PCT. Similarly, small peptide sequences that display high affinity towards PCT may represent the future of PCT sensors. The emerging technologies such as 3D printing, microfluidics, and machine learning could further improve the clinical potential of point-of-care devices for determination of PCT. Nevertheless, in order to fabricate superior performance PCT sensors in futuristic point-of-view, concerted efforts are needed for identification of the ideal combination of nanointerface materials, understanding the reactions occurring at the electrode-electrolyte interface and the sensing mechanisms involved. Multiplex detection of PCT along with other disease markers such as C-reactive protein (CRP), and pro-inflammatory cytokines such as IL-8, IL-6 and TNF$\alpha$, could be coupled for accurate and early clinical diagnosis of many inflammation-mediated disorders.

**Author Contributions:** S.N. collected the literatures and prepared the original draft of the manuscript. A.P. and P.K.M. were involved in compiling the figures and formatting the references. U.M.K. conceptualized the idea and corrected the manuscript. All authors have read and agreed to the published version of the manuscript.

**Funding:** This work was supported by RCUK (Grant number: MR/P027881/1).

**Institutional Review Board Statement:** Not applicable.

**Informed Consent Statement:** Not applicable.

**Data Availability Statement:** Not applicable.

**Acknowledgments:** The authors would like to acknowledge SASTRA Deemed University for infrastructural support and RCUK (Grant number: MR/P027881/1) for financial support.

**Conflicts of Interest:** The authors declare no competing financial interest.

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
