# Peer review of "Electrochemical Immunosensors for Quantification of Procalcitonin: Progress and Prospects"

_chemosensors, doi:10.3390/chemosensors9070182_

Round 1

Reviewer 1 Report

Dear Editor,

The manuscript entitled “Electrochemical Immunosensors for Quantification of Procalcitonin: Progress and Prospects” by Nellaiappan et al. reviews the existing detection strategies and limitations of human procalcitonin (PCT) electrochemical immunosensors. The authors summarize the available electrochemical immunosensors for sensitive and selective recognition of PCT in serum samples and the PCT immunosensing platforms are classified by their mode of signal transduction (either labelled or label-free), signal amplifying materials and signal tagging elements.

In my opinion, the manuscripts’ objective and perspective are very interesting, the manuscript is well-written and should be accepted for publication after minor revisions. My detailed comments for the authors to consider are provided below:

  1. A graphic describing the direct and sandwich detection principles would be for the reader.
  2. The respective references should be included in the introduction paragraph after each used material even though they are included in table 1.
  3. A small sentence with electrochemical principles description would be useful in the introduction section.
  4. Reference 45 misses its authors.

Overall, it is a very well written and informative manuscript, in my opinion.

Reviewer 2 Report

In this work, authors reviewed electrochemical immunosensors for the analysis of human procalcitonin, a biomarker of infections and sepsis. The topic is interesting and authors reviewed a significant number of related works.

Some comments:

- Rewrite the sentence “Another popular strategy involved in PCT detection appears to involve the use of gold labelled immune complexes for generating the electrochemical or optical response”.  Since in the previous sentence it is said that “the most extensively investigated transduction mechanisms for quantification of PCT has been optical and electrochemical methods”, the generation of electrochemical or optical response is not “another strategy”.

- Rewrite the sentence “Recently, nanomaterials have elicited significant interest due to the…”. Nanomaterials have been used for improve the signal in electrochemical sensors for long time.

- In sentence “This is because the sandwich approach provides signal amplification and more specificity”, explain how the sandwich immunosensors provide signal amplification.

- Include references in the sentences “Various immunosensor platforms have been utilized such as nanometals (gold, platinum, zinc), metal oxides (cerium oxide, molybdenum oxide), inorganic complexes (cobalt phthalocyanine), carbon nanomaterials (graphene, reduced graphene oxide, multi-walled/single-walled carbon nanotubes, ordered mesoporous carbon, fullerene C60), quantum dots (zinc sulfide capped cadmium selenide) and highly branched polymers (poly(amidoamine), PAMAM) with traditional redox partners (ferrocene, thionine and toluidine blue)”; and “Both layer-by-layer modification or sandwich type arrangement coupled with either labelled or label/enzyme-free immunoassays have been explored for detection of PCT”.

- Place the table 1 closer to the paragraph in which it is first mentioned.

- In the sentence “The performance and biocompatibility of the existing electrochemical immunosensors are summarized and tabulated in Table 1”, how the biocompatibility is summarized in that table? Explain what you mean.

- Explain better the references 19, 21 and 26.

- Improve the conclusion section: it shouldn’t include what is the review about, but it should summarize the best strategies for developing immunosensors for this biomarker, and the future trends regarding electrochemical immunosensors for clinical applications, and specifically for this biomarker.

Reviewer 3 Report

This article reviews the applications of electrochemical immunosensors to the determination of procalcitonin, a biomarker of various diseases. In my opinion, the article is of relative interest to researchers working in this field, since the number of designs already described in the literature is very high, and new research is probably directed at other emerging biomarkers of greater importance today. However, the amount of information provided in the paper is high and the manuscript is well written. From my point of view, the authors should have provided more criticism, highlighring the advantages of these immunosensors compared to other types based on optical transducers, or ELISA-type immunoassay or genosensors. Furthermore, the overly basic information such as the description of the types of immunosensors or the foundation of the sandwich type configuration, should be removed.

Reviewer 4 Report

The review from Nellaiappan presents an interesting overview in the field of Electrochemical Immunosensors for Quantification of Procalcitonin. The topic is certainly worth investigating and in general the authors’ efforts have resulted in a nice work which would suitable for publication given these revisions to improve the quality of the manuscript:

    1. In the introduction, the authors should briefly mention the reader that there exist optical sensors for Procalcitonin for instance based on convenient colorimetric detection (e.g. Sensors and Actuators B: Chemical, 316, 1 August 2020, 128163). Accordingly, the authors should clearly report the advantages of electrochemical detection approaches vs. optical ones.
    2. Please include in the introduction the time interval considered for the literature critical search. Did the authors look into papers in the last five years or last decade? Please specify.
    3. A simple scheme reporting the different methods for sensing (sandwich type, label-free, enzyme-free) is needed to make the reader aware of the pros and cons of each approach.
    4. It would be important to cite the minireview (Int. J. Electrochem. Sci., 15, 2020, 6436 – 6447, doi: 10.20964/2020.07.47) which has a similar topic to the one herein explored, in order to clearly ascertain the differences.
    5. Page 3: Please check the meaning of this sentence: “It is evident that the need of the hour is a simple, rapid and sensitive detection strategy for PCT that also offers a wide quantification range.”
    6. Page 16: “The removal of the qantum dots” should be written as “The removal of the quantum dots”
    7. Did the authors find some possible applications sensing applications involving DNA or RNA aptamers for Procalcitonin?
    8. I believe that an example of microfluidic sensor should be inserted, as the one reported in” “Analyst, 2020,145, 5004-5010, doi: 10.1039/D0AN00624F”.

Round 2

Reviewer 2 Report

The article has been improved.

Reviewer 4 Report

The authors have sufficiently responded to all the raised observations and, in doing so, they have improved the overall quality of their manuscript.